# Therapeutic Approach of Whole-Body Vibration Exercise on Wound Healing in Animal Models: A Systematic Review

**DOI:** 10.3390/ijerph20064925

**Published:** 2023-03-10

**Authors:** Adrielli Brites-Ferreira, Redha Taiar, André Luiz Bandeira Dionizio Cardoso, Daysa De Souza-Santos, Patricia Prado da Costa-Borges, Luiza Torres-Nunes, Luelia Teles Jaques-Albuquerque, Bruno Bessa Monteiro-Oliveira, Francois Constant Boyer, Danúbia da Cunha Sá-Caputo, Amandine Rapin, Mario Bernardo-Filho

**Affiliations:** 1Programa de Pós-Graduação em Fisiopatologia Clínica e Experimental, Faculdade de Ciências Médicas, Universidade do Estado do Rio de Janeiro, Rio de Janeiro 20551-030, Brazil; 2Laboratório de Vibrações Mecânicas e Práticas Integrativas—LAVIMPI, Departamento de Biofísica e Biometria, Instituto de Biologia Roberto Alcantara Gomes and Policlínica Universitária Piquet Carneiro, Universidade do Estado do Rio de Janeiro, Rio de Janeiro 20950-003, Brazil; 3MATériaux et Ingénierie Mécanique (MATIM), Université de Reims, 51100 Reims, France; 4Programa de Pós-Graduação em Saúde, Medicina Laboratorial e Tecnologia Forense, Universidade do Estado do Rio de Janeiro, Rio de Janeiro 20950-003, Brazil; 5Centre Hospitalo-Universitaire de Reims, Département de Médecine Physique et de Réadaptation, Hôpital Sébastopol, Université de Reims Champagne-Ardenne, 51092 Reims, France; 6Faculté de Médecine, VieFra, Université de Reims Champagne-Ardenne, 51097 Reims, France

**Keywords:** vibration therapy, animal studies, wound, healing, skin

## Abstract

Human skin wounds pose a gathering threat to the public health, carrying an immense epidemiologic and financial burden. Pharmacological and non-pharmacological (NP) treatments have been proposed to the management of wound healing. Physical exercise is a strong NP intervention considered for patients in wound healing. Particularly, a type of exercise intervention known as whole-body vibration (WBV) exercise has gained increasing interest. WBV exercise is generated due to the transmission of mechanical vibrations, produced by a vibrating platform, to the body. The aim of this review was to summarize studies in experimental animal models using WBV exercise in wound healing. Searches were performed in EMBASE, PubMed, Scopus and Web of Science including publications on 21 November 2022 using the string “whole body vibration” AND “wound healing” (animal or mice or mouse or rat or rodent). The SYRCLE tool was used to assess the risk of bias (RoB). From 48 studies, five studies met the inclusion criteria. RoB indicated that none of the studies fulfilled all methodological analyzed criteria, resulting in possible biases. The studies were homogeneous, and results suggest beneficial effects of WBV exercise in wound healing, mainly related to enhancing angiogenesis, granulation tissue formation, reducing the blood glucose level and enhancing blood microcirculation, by increasing myofiber growth and rapid re-epithelialization. In conclusion, the various biological effects of the response to the WBV exercise indicate the relevance of this intervention in wound healing in animals. Moreover, considering the translation approach, it is possible to speculate that the beneficial effects of this non-pharmacological therapy might justify clinical trials for wound healing also in humans, after criterion evaluation.

## 1. Introduction

Human skin wounds represent a gathering threat to the public health and carry an immense epidemiologic and financial burden, affecting a large fraction of the world population annually [1]. In developed countries, it has been estimated that 1–2% of the population experience a skin wound during their lifetime [2]. In addition, the associated costs projections ranged from $28.1 billion to $96.8 billion of the total health care expenses [3].

In healthy individuals, restoration of a functional epidermal barrier is highly efficient [4]. However, when the normal repair response goes awry, there are two major outcomes: either an ulcerative skin defect (chronic wound) or excessive formation of scar tissue (hypertrophic scar or keloid) [5]. In fact, wound healing is an important process which contains three stages regulated by sequential yet overlapping phases such as the hemostasis/inflammation phase, the proliferation phase, and the remodeling phase [6]. Thus, an important interplay between many cellular skin agents, such as keratinocytes, fibroblasts, endothelial cells of vessels and recruited immune cells, and those associated with the extracellular matrix is observed [7]. In contrast, chronic wounds most often do not progress past the inflammatory phase [8].

Chronic wounds have become a major challenge to healthcare systems worldwide. They are classified into four categories such as arterial, diabetic, pressure and venous ulcers [8]. This impaired healing generally affects the adult population and is growing more associated with the elderly, increasing with progressively prevalent disease states such as diabetes mellitus, vascular diseases and obesity [9]. Health professionals, particularly those in surgical specialties and who deal with healing disorders daily, have observed chronic non-healing wounds increasing such as diabetic foot ulcers (DFUs), venous leg ulcers (VLUs) and pressure ulcers (Pus) [10]. In this manner, new therapeutic approaches and the continuing development of technologies for acute and long-term wound management have been proposed [11].

Management practices on wound healing include pharmacological and non-pharmacological treatment. Pharmacological includes (i) treatment with anti-microbial and (ii) anti-inflammatory drugs [12]. In turn, non-pharmacological treatment includes: (i) debridement (including surgery), (ii) offloading (or compression), (iii) management of ischemia, (iv) management of infection, by mechanical and electrical stimulation, (vi) appropriate wound bed preparation and dressing [8,11,13,14] and (vii) guidance to improve lifestyle, including exercise and cessation of smoking [15], thorough assessment with medical and nutritional optimization [16].

Accordingly, whole-body vibration (WBV) exercise, as systemic vibratory therapy [17], has been proposed as an alternative exercise intervention for certain populations who are not able to perform adequate active exercise trainings [18]. WBV is a clinical intervention in which subjects are exposed to mechanical vibration (MV) generated in a vibrating platform [19]. In the protocols of WBV exercise, biomechanical parameters, such as the frequency, amplitude, the peak-to-peak displacement and peak acceleration, must be defined, besides the work time and rest time among the sessions [20].

Studies have shown that low-intensity vibration (LIV) can improve angiogenesis and wound healing in diabetic mice, potentially by increasing growth factors such as insulin-like growth factor (IGF)-1 and vascular endothelial growth factor (VEGF) in the wound [21]. Moreover, low magnitude high frequency vibration (LMHFV) has been proven as an alternative to conventional exercise that can stimulate metabolic responses [22]. In preclinical and clinical studies, LMHFV has been reported to have benefits on muscle contractibility functions [23], muscle structures and strength [24]. It has also been proven to be effective in inducing tissue regeneration and promoting blood flow [25].

WBV exercise has acquired acceptance in a number of clinical contexts including physical disability [26], sports rehabilitation [27], body composition [28], multiple sclerosis [29], osteoporosis [30], lumbar disk disease and lower back pain [26], cerebral palsy [31] and mental health [32,33]. Also, it has been used to improving functional mobility and neuromuscular performance in different patient groups [10,30] and athletes [34], including endurance, power and muscle strength [35].

WBV exercise has been proved to be an applicable intervention under different conditions [17,36]. Additionally, it can be applied to (small) animals with strong translational purpose [36]. The potential physiological effects of WBV on different organs/tissues would be related to possible neuromuscular responses and the tonic vibration reflex [19,20,37]. Moreover, there is a mechano-transduction mechanism in which body cells, once stimulated by the mechanical vibration, modulate biological activity through specific signaling pathways, such as releasing hormones and other substances (e.g., amino acids, proteins, lipids, ions) [17]. In turn, experimental models with various approaches were considered relevant to try comprehending the effects of WBV exposure. Therefore, the purpose of current systematic review is to summarize findings of experimental animal models related to effects of the WBV in wound healing. In addition, the findings might reinforce the efficiency and safety of mechanical vibration treatment in improving angiogenesis, increasing collagen deposition and in pro-angiogenic growth factors and accelerating the process of wound healing, which may serve as a reference for the design of new clinical trials.

## 2. Materials and Methods

### 2.1. Search Strategy

Guidelines of the Preferred Reporting Items for Systematic Reviews and Meta-Analyses [PRISMA statement] [38] will follow to write the current systematic review. The search about this systematic review in the International Prospective Register of Systematic Reviews (PROSPERO) revealed that there was not review in this subject. The submission protocol of the current systematic in the PROSPERO is 403483. Searches in electronic databases were carried out to identify and select only animal studies that used WBV. The searches were performed on PubMed/MEDLINE, Embase, Scopus and Web of Science (on 21 November 2022), using the string “whole body vibration” AND “wound healing” (animal or mice or mouse or rat or rodent). No publication date restriction was imposed.

“Does WBV exposure accelerates wound healing in animal models?” was the guided question of this systematic review. PICO tool was used [39], and it was considered, P (Population)—experimental animal models and any method of inducing a wound model; I (Intervention)—WBV; C (Comparison)—other treatments or no comparison; and O (Outcome)—comprehension about what effects WBV exposure can promote in wound healing recovery time in experimental models.

### 2.2. Inclusion and Exclusion Criteria

The selection of the studies was performed independently by two reviewers (A.B.F and D.S) considering the inclusion criteria: (1) animal studies, (2) use of WBV for fundamental research or therapeutic purpose, (3) in vitro trials or in vivo measurements that investigated the effects of MV on wound healing, (4) evaluation of short and/or long-term intervention and (5) full-text article published in English.

Exclusion criteria of the studies were (1) short communications, (2) review articles, (3) case reports, (4) books, (5) consensus statements, (6) expert opinion, (7) articles published in other languages (e.g., Spanish, Chinese, Russian, Arabic), (8) publications that used WBV associated with other product or therapy and (9) therapy used in humans.

### 2.3. Data Extraction

Data were obtained from the full-text version of the publications by three reviewers (A.B.F, D.S and L.JA). Data included the year of publication, authors, species, aim, intervention protocol, WBV intervention, study design and results. The extracted data were analyzed, and discrepancies were discussed with a fourth reviewer (D.S.P).

### 2.4. Appraisal of Risk of Bias (RoB)

Independently, two reviewers (A.B.F and A.C) estimated the RoB of the selected studies and the Systematic Review Centre for Laboratory Animal Experimentation (SYRCLE) protocol was followed. This was an adapted version of Hooijmans et al. [40], in which, in case of missing information, an unclear RoB is scored instead of a low RoB. The conflicts were solved by a fourth reviewer (M.B).

## 3. Results

According to the search strategy performed in electronic databases, a total of 48 articles were identified: 24 articles from PubMed, seven from Scopus, 12 from Embase and five from Web of Science, containing articles from 2010 to 2022. After removing duplicates (15 articles), 33 articles were screened, and the abstracts reviewed. Of those, 26 articles were excluded because the abstracts showed that they did not match the inclusion criteria. Thus, seven articles were identified as potentially eligible and subjected to more detailed analysis. Two were eliminated for the following reasons: (i) study about the effect of WBV in decreasing pain in patients with burns to the extremities and (ii) a study about bone fracture healing (Figure 1).

### 3.1. Selected Studies

As observed in Table 1, the selected articles (n = 5) were organized by year of publication, authors, animal species, aim, intervention protocol, WBV intervention, study design and results (Table 1). All studies (100%, n = 5) involved the WBV tests in wound models; a large percentage of studies (60%, n = 3) [21,41,42] were designed evaluating, in animals, clinical condition such as diabetes; one study (20%, n = 1) [43] looked at wound healing driven by pressure ulcers; and one study (20%, n = 1) [44] used LIV for muscle healing after trauma injury. Likewise, in all studies wounds were induced, and after induction, the study started.

Mice (*Mus musculus*) were the main animal model used on selected studies (80%, n = 4) [21,41,43,44], following by rats (*Rattus norvegicus*, 20%, n = 1) [42]. Among the animal models used in the studies, 80% (n = 4) [21,41,43,44] of the animals were males, and 20% (n = 1) [42] were female. Moreover, in regard to mice strains, each study used a different strain, where it was used mice db /db (40%, n = 2) [21,41], followed by ICR (20%, n = 1) [43]. Among the rat strain study, *Wistar* rats (20%, n = 1) [42] were selected. All studies reported the age of experimental animals, which ranged from 5 days to 13 weeks. About the animals’ body mass, only one study using the mice model reported a body mass of 35–40 g.

Many vibration protocols are found in the included articles, with frequencies ranging from 35 to 90 Hz and peak acceleration ranging from 0.2 a 0.6 g. Two studies [41,44] reported a comparison of the effects of different frequencies, and one study makes four types of comparisons of high and low frequency, with high and low magnitude vibration.

The timing of implementation of the WBV protocol varied among the studies, ranging from 7 to 15 days after wound induction. In addition, 100% (five studies) of the protocols used only one instance of treatment per day with the duration ranging from 20 to 30 min/ day. Two studies (40% = 2) [21,43] subjected mice to vibration for 5 consecutive days, for 30 min each instance of treatment. One study (20% = 1) [42] subjected the rat to vibration for 5 consecutive days for 20 min each instance of treatment, and two studies (40% = 2) [41,44] subjected mice to vibration for 7 consecutive days, for 30 min for each instance of treatment.

### 3.2. Main Findings

A large number of findings considered favorable for WBV application were observed in the selected studies, as summarized in Table 1. All studies resulted in improved healing. In studies of diabetic mice, the WBV application increased angiogenesis and granulation tissue formation (n = 3) [21,41,42]; surface measurements revealed a trend towards accelerated closure in wounds from WBV-treated mice. The studies observed a change in the accumulation of inflammatory cells in the wound, with an increase significant in VEGF levels.

Only one study reported an improvement in blood microcirculation, which justified the increase in cell proliferation, angiogenesis and local inflammatory response. This same study was also the only one that reported a reduction on blood glucose level; the flux data further supported that WBV enhances glucose uptake after increasing blood flow [35,42].

Corbiere et al. [44] evaluated the effect of WBV on muscle healing with two different protocols. The two protocols (90 Hz/0.2 g and 45 Hz/0.4 g) improved the healing of lacerated gastrocnemius for a time of 14 days post-injury; there was a trend towards an increase in percent area of peripherally nucleated fibers in the groups treated with WBV. However, only the 45 Hz/0.4 g protocol showed an improvement in the muscle healing by enhancing myofiber growth and reducing fibrosis. Wano et al. [43] assessed effects of WBV on wound healing in a mouse pressure ulcer model and observed a significant decrease in neutrophil infiltration and TNF-α levels in wounds.

Regarding to the experiment period, two studies observed that wound closure rate did not differ between the groups (control and WBV) on day 7 post-ulceration; the reduction in wound size was only noticed on day 14, following WBV treatment [43,44]. One of these studies used the WBV protocol with 90 Hz and 0.2 g for a seven-day treatment period and did [44] not noticeably improve the healing of lacerated gastrocnemius muscle, and no differences were found in markers for angiogenesis or macrophage accumulation. This data indicates that seven days of WBV treatment may not be sufficient to induce observable improvements in the healing process.

No harmful effects to the animals were reported due to the WBV exposure in the selected publications.

### 3.3. Risk of Bias

The quality assessment is shown in Figure 2. No studies fulfilled all methodological analyzed criteria. Concerning the selection bias, the sequence generation process was not fully reported in 20% (n = 1) (Q1). In terms of the characteristics of the animals, that is, the similarity to one another (Q2), 20% of the studies (n = 1) did not report this information clearly. Information about the allocation concealment (Q3) was not clear in all studies. Only 40% (n = 2) of the studies reported a random animal housing (Q4). In 20% (n = 1), blinding of personnel was described, while 60% (n = 3) were unclear, and 20% (n = 1) did not report blinding of personnel (Q5). Random selection for outcome assessment (Q6) was applied to 20% (n = 1). Only 20% (n = 1) of the studies reported blinding of outcome assessment (Q7); 40% (n = 2) had unclear information. Incomplete outcome data (Q8) were addressed as unclear by the reviewers because all the studies did not meet all questions from this topic.

Selective reporting (Q9) shows a low risk of the studies. An important source of bias was the lack of information about animal models’ age, such as the body mass of the mice species used and the number of animals that was not specified in 20% (n = 2) (Q10). Considering each criterion analyzed individually, none of the studies reported whether the allocation to the different groups was adequately concealed during the study. In addition, they did not report whether the incomplete outcome data were adequately addressed and details of the sample size calculation as well, which made it unclear how to decide whether it was a low or high RoB. Additionally, the analysis of the individual studies found a possible relation between RoB and year of publication, as shown in Figure 3.

## 4. Discussion

The current study aimed to summarize, in experimental animal models, impacts of the WBV exposure on wound healing. Thus, it might help to understand the effective training parameters, considering its translational relevance for wound treatment in clinical research. This systematic review included five studies discussing effects of WBV on wound healing process.

Despite the homogeneity of the studies included in this current review, the WBV intervention varied in duration, in periodicity of the interventions, session duration and biomechanical parameters (frequency, amplitude). The preliminary results suggest beneficial effects of the WBV intervention on wound healing. Enhancement of angiogenesis, granulation tissue formation, reduction of blood glucose level, enhancement of blood microcirculation by increasing myofiber growth and reducing fibrosis and a decrease in TNF-α levels and neutrophil infiltration by collagen deposition and rapid re-epithelialization were found. No harmful effects to the animals were reported due to the WBV exposure, and this suggests that this intervention is safe with controlled parameters.

Considering the WBV protocols described in the selected publications, there were differences about the device used to generate MV and the used parameters. Some common features are presented in Table 1. Considering biomechanical parameters, the most used MV frequencies were 45 and 90 Hz, and magnitudes were from 0.3 to 0.6 g. Considering the sessions of the intervention protocol, most protocols involved five or more interventions sessions per week. The outcome of the treatment protocol was similar among the studies; Rita et al. [41] compared two different protocols, and Corbiere et al. [44] compared four protocols; both had similar results to the other groups. Protocols with low intensity and low magnitude vibration (45 Hz, 0.3 or 0.4 g) had a better healing response compared to protocols with high frequency. The protocol described by Corbiere et al. [44] involved low-frequency, high-magnitude, high-frequency and high-magnitude vibration that impaired the wound closure. Moreover, a study demonstrated that LIV intervention with similar frequency (47 Hz) and acceleration (0.2 g) on pressure ulcers (3.15 min treatments per day) can improve healing of stage I pressure ulcers in elderly patients compared to standard care [45].

In a study, a high-intensity WBV protocol (90 Hz and 0.15 g) to improve bone loss in rodents was used [46]. It was verified that the protocol was able to stimulate bone anabolic activity, suggesting that WBV can act as an anabolic signal to a skeleton even upon the withdrawal of estrogen [46]. However, Stuermer et al. [47] observed that the vibration intervention seemed to be unfavorable for fracture healing. Gnyubkin et al. [48] reported positive results with high-intensity protocols. There is the promotion, in three weeks of WBV, an acceleration of cortical bone growth and osteoclast surfaces, which after nine weeks of WBV result in a net increase in bone matrix mineral content.

A robust increase in angiogenesis was reported in three studies from day 7, 8 and 10. In two studies [43,44], a relevant reduction in wound size and a significant increase in collagen deposition particularly on day 14, following WBV treatment, were reported.

Concerning the main results of the current systematic review, WBV intervention was responsible for several responses such as wound healing by granulation tissue formation, re-epithelialization and angiogenesis. Comparing the selected five studies, only the study by Wano et al. [43] was controversial on the effect of WBV in increasing the angiogenesis. Diminished production of pro-angiogenic growth factors, such as IGF-1 and VEGF, is thought to contribute to the impaired angiogenesis in chronic wounds associated with diabetes [49]. It is suggested that LIV may exert these pro-angiogenic effects via the actions of IGF-1 and VEGF [21]. VEGF can induce angiogenesis by increasing endothelial cell migration and proliferation [50]. IGF-1 would induce endothelial cell migration via chemotactic activity in endothelial cell lines [51]. Macrophages are also broadly connected with angiogenesis and healing [52,53].

It is relevant to consider that LIV can be anabolic to bone and that the mechanical signals do not need to be of an important magnitude to elicit an anabolic effect. Thus, LIV may promote the mobilization and/or homing of bone marrow-derived cells to the damaged tissue. These cells would include progenitor cells and monocytes/macrophages, which have important role in the tissue repair as growth factors and cytokines that promote tissue healing released [44,54,55].

It is known that cells, such as monocytes/macrophages, fibroblasts and/or endothelial cells, promote wound healing in part by producing growth factors that induce angiogenesis, collagen deposition and wound closure [56,57]. LIV leads in a trend towards a less inflammatory phenotype in CD11b2 cells but not in CD11b+ macrophages in the wound, and it is suggested that LIV may exert pro-angiogenic effects via cells such as fibroblasts, endothelial cells and/or keratinocytes [21].

Games et al. [55] reported that WBV can increase skin blood flow in both rodents and humans, which could impact the wound healing along with other physiological responses [58]. Wano et al. [43] propose that the vasodilating WBV effect is responsible for enhancing blood supply to the wound sites, which provide tissue oxygenation, and this contributes to wound repair. Moreover, Murray et al. [59], reported that the rise in blood flow justifies the formation of granulation. This would be related to blood microcirculation, since granulated regions have a greater blood flow than those without granulation. This is consistent with the results reported by studies included in the current review.

Corbiere et al. [44] observed that MV improved muscle healing in a mouse model of laceration injury, and the obtained results were similar in relation to skin healing. Two protocols, 90 Hz and 0.2 g and 45 Hz and 0.4 g for 14 days were used. Both protocols improved the healing of lacerated gastrocnemius muscle and increased myofiber growth after formation. However, only with the 45 Hz and 0.4 g protocol was there observed an improvement in the reduction of fibrosis. These findings justify the hypothesis that protocols with low intensities are more favorable for healing. The results suggest that LIV may have anabolic effects on regenerating muscle and the elucidating mechanisms underlying the local and/or systemic effects of LIV. This would be explained due to muscle being particularly sensitive to mechanical stimuli or indirectly through the production of cytokines and growth factors that promote muscle growth.

The current systematic review has some limitations. The search strategy used only four electronic databases. Moreover, only publications in English language were searched to be included. The RoB and methodological quality assessment showed that studies failed to establish a consistent method to avoid divergences, leading to possible biases.

Then, it is possible to endorse, as a fact and scientific contribution, that WBV exercise is useful to treat chronic wounds in animals, and, considering the translational approach, this non-pharmacological intervention would be a perspective for a clinical application in human beings. Moreover, the findings noticed in this review are relevant to support the formulation of appropriate protocols in clinical studies, considering the benefits and biological effects of the WBV exercise.

## 5. Conclusions

In conclusion, putting together all the findings that were reported in the current systematic review, it is possible to consider that WBV might be a feasible and effective therapy for the treatment of wound chronic in animals. The reasons would be: (i) an adequate knowledge about the WBV parameters to be used in the protocols, avoiding undesirable effects, (ii) a possible WBV modulating action verified at TNF-α levels and neutrophil infiltration, (iii) improvement of the blood microcirculation, by increasing myofiber growth and reducing fibrosis and (iv) by enhancing angiogenesis. Considering the translation approach, WBV may represent an attractive supporting alternative for clinical studies.

## Figures and Tables

**Figure 1 ijerph-20-04925-f001:**
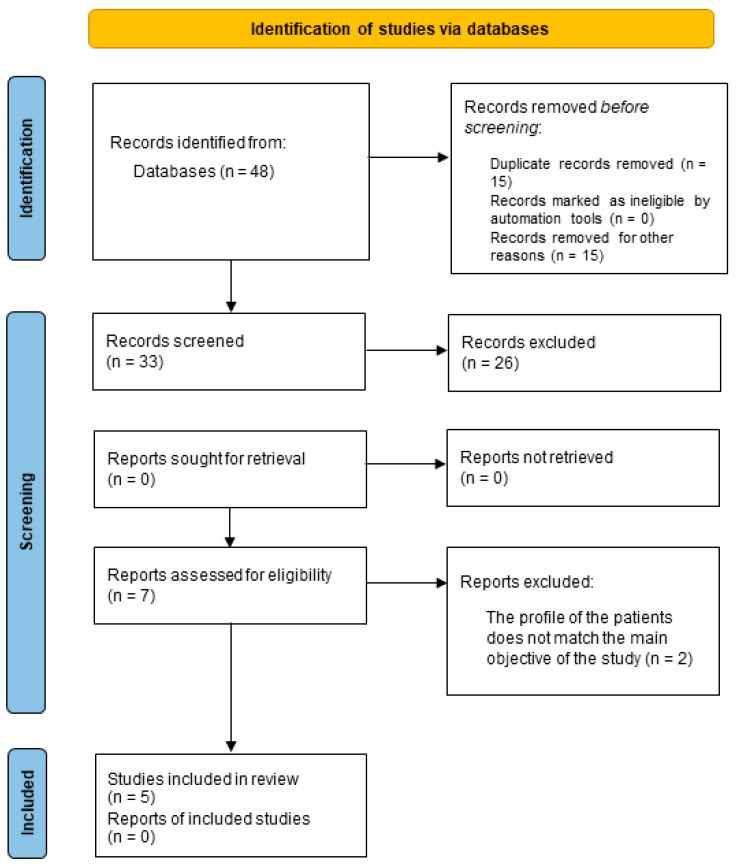
Flow chart of study selection process based on PRISMA guidelines.

**Figure 2 ijerph-20-04925-f002:**
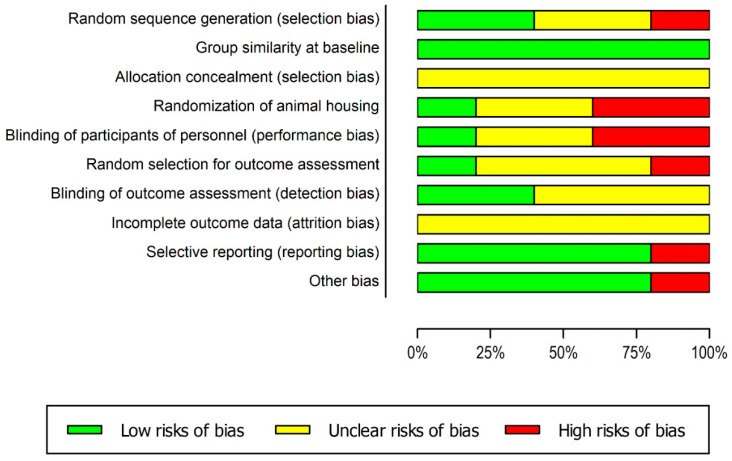
Results of the risk of bias and methodological quality indicators for all included studies in this systematic review that evaluated the effect of WBV treatment on brain and behavior in experimental models. The items in the Systematic Review Centre for Laboratory Animal Experimentation (SYRCLE). Risk of Bias assessment were scored with ‘yes’ indicating low risk of bias, ‘no’ indicating high risk of bias or ‘unclear’ indicating that the item was not reported, resulting in an unknown risk of bias.

**Figure 3 ijerph-20-04925-f003:**
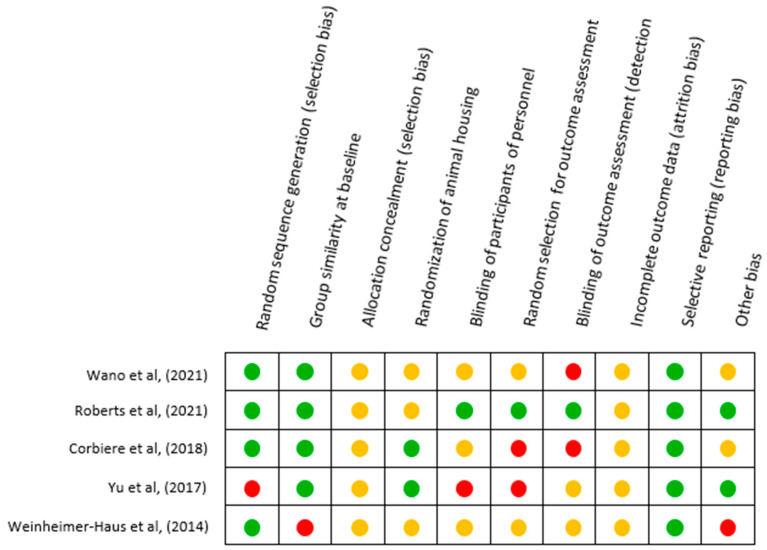
Risk of bias summary shows studies’ quality assessment at an individual level [21,41,42,43,44]. (green) Low risk of bias. (red) High risk of bias. (yellow) Unclear risk of bias.

**Table 1 ijerph-20-04925-t001:** Summary of in vivo experimental studies investigating the effects of WBV in wound healing.

Authors	Species	Aim	WBV Intervention	Period (Days a Week) Duration of Bouts	Study Design	Results
Weinheimer-Haus et al., (2014) [21]	Male Mice db/db (N = Undefined)	The objective was to assess effects of whole-body LIV on wound healing in diabetic db/db mice.	LIV: Vertically F: 45 Hz apeak of: 0.4 g	2-Weeks (5 days) 1 bout: 30 min	- WBV group: 7 days (n = ± 13/16) - CON: 7 days (n = ±13/16) - WBV group: 15 days (n = ±8/9) - CON: 15 days (n = ±8/9)	LIV may exert beneficial effects accelerating wound closure by enhancing angiogenesis and granulation tissue formation, and these changes are associated with increases in pro-angiogenic growth factors. Wound closure was significantly higher in the WBV group compared to the CON group (*p* = 0.066).
Yu et al., (2017) [42]	Female *Wistar* Rats (N = 96)	The objective was to investigate effects of LIV on the DM at Days 1, 4, 8 and 13 post-wounding.	LIV: F: 35 Hz /apeak: 0.3 g	2-Weeks (5 days) 1 bout: 20 min	- DM_V (n = 24) - DM (n = 24) - Ctrl_V (n = 24) - Ctrl (n = 24)	LIV accelerates the foot wound healing; in DM_V and DM, there was reduction in the wound size; in DM-V, there was reduction in the blood glucose level and the glucose transporter 4 expression and enhanced blood microcirculation. Wound size decreased significantly from day 8, there was a significant difference between the DM_V group compared to the control group (*p* = 0.036).
Corbiere et al., (2018) [44]	Male C57BL/6J Mice (N = 78)	The objective was sought to determine whether mechanical stimulation via LIV could improve muscle healing following traumatic injury	LIV: F: 90 Hz apeak: 0.2 g or F: 45 Hz apeak: 0.4 g	2-Weeks (7 days) 1 bout 30 min	- LIV 90 Hz, 14 days (n = 14) - LIV 45 Hz, 14 days (n = 18) - LIV 90 Hz, 7 days (n = 6) - CON (n = 16; n = 18; and n = 6)	LIV can improve muscle healing by increasing myofiber growth and reducing fibrosis. Statistical differences were observed in LIV groups compared to the control group (*p* ≤ 0.05).
Wano et al., (2021) [43]	Male ICR Mice (N = 32)	The objective was to examine effects of WBV on the healing of stage II pressure ulcers in a mouse model.	LIV: Vertically F: 45 Hz apeak: 0.4 g	2-Weeks (5 days) 1 bout: 30 min	- WBV (n = 16) - CON (n = 16)	TNF-α levels and neutrophil infiltration were significantly decreased in wounds on days 7 and 14 of WBV treatment; wound closure rate and collagen deposition were remarkably accelerated on day 14. On day 14, the WBV group showed an improvement in healing compared to the control group (*p* < 0.01).
Roberts et al., (2021) [41]	Male db/db Mice (N = Undefined)	The objective was to identify LIV amplitudes and frequencies that promote healing in diabetic mice.	LIV: F: 45 and 90 Hz apeak: 0.3 g and 0.6 g	2-Weeks (7 days) 1 bout: 30 min	- LL 0.3 g, 45 Hz - LH 0.6 g, 45 Hz - HL 0.3 g, 90 Hz - HH 0.6 g, 90 Hz - CON	The 45 Hz/0.3 g group was the only one that improved wound healing and increased angiogenesis and granulation tissue formation, rapid re-epithelialization and wound closure. The LL group had a mean value significantly different from the control group (*p* ≤ 0.05).

F: frequency (Hz); apeak: peak acceleration in multiple of Earth’s gravity (g); n: number; LIV: Low-Intensity Vibration; DM: Diabetes Mellitus; WBV: Whole Body Vibration; T: Time; I: Intervention; DM_V: DM-Vibrated; DM: DM-Control; Ctrl_V = Non-DM Vibrated; Ctrl: Non-DM Control; CON: Control group; LL: low acceleration, low frequency; LH: low acceleration, high frequency; HL: high acceleration, low frequency; HH: high acceleration, high frequency; ICR: Institute of Cancer Research.

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
