# Peer review of "Therapeutic Approach of Whole-Body Vibration Exercise on Wound Healing in Animal Models: A Systematic Review"

_ijerph, 2023, doi:10.3390/ijerph20064925_

Round 1

Reviewer 1 Report

This manuscript is clinically relevant and authors defined PICO of the study adequately. However, some corrections should be addressed:

(1) There is not registration number of this systematic review and was not added on the PROSPERO website. Please check carefully this issue.

(2) Authors must consider to extend terms MESH related to whole body vibration (i.e., “WBV” or “side alternating vibration” or “vertical vibration”, etc.) in order to include more studies in this systematic review.

(3) Table 1 is not clear. Authors should include an effect estimate and its precision (e.g. confidence interval, mean, standard deviation, p-value between pre-post interventions) in results column.

(4) Be sure the references in-text citations follows Vancouver format.

Author Response

Manuscript Minor Revision

Reviewer #1: IJERPH-2210179

This manuscript is clinically relevant and authors defined PICO of the study adequately. However, some corrections should be addressed:

(1) There is not registration number of this systematic review and was not added on the PROSPERO website. Please check carefully this issue.

Answer - We thank the reviewer for the positive remarks and the constructive contributions to our manuscript. We added the information in the Material and Methods section “The search about this systematic review in the International Prospective Register of Systematic Reviews (PROSPERO) revealed that there was not review in this subject. The sub-mission protocol of the current systematic in the PROSPERO is 403483.”

(2) Authors must consider to extend terms MESH related to whole body vibration (i.e., “WBV” or “side alternating vibration” or “vertical vibration”, etc.) in order to include more studies in this systematic review.

Answer - We thank the reviewer for the constructive remark. We did a search in the selected databases with this MESH terms suggested by the reviewer, but we found the same records that were first identify in the search made with the originally search terms proposed for this study. Therefore, none of them increase our search results regard to screening, specifically in eligibility section and in final inclusion.

(3) Table 1 is not clear. Authors should include an effect estimate and its precision (e.g. confidence interval, mean, standard deviation, p-value between pre-post interventions) in results column.

Answer - We thank the reviewer for the positive contributions to our analysis. We added the p-value among interventions in the column "RESULTS", as shown in the table 1.

(4) Be sure the references in-text citations follows Vancouver format.

Answer - We thank the reviewer for the positive contribution. We added the square brackets around all citations in the text according to this journal instructions for references.

Reviewer 2 Report

please see attached

Author Response

Manuscript Minor Revision

Reviewer #2: IJERPH-2210179

The abstract is accurate. The SYRCLE tool that was used to assess the risk of bias is very good for animal studies. I would write it as RoB (line 36)

Answer - We thank the reviewer for the positive remark. We followed your suggestion, and we made this change in the abstract.

Introduction

The introduction is fine, but you NEED to put your in-text citations in brackets as it is suggested by the journal.

- Line 74-75: re-write that part. I guess you mean the management of infection with the use of mechanical and electrical stimulation.

Answer - We appreciated your consideration, so we rewrote this sentence in the fourth paragraph of the “INTRODUCTION” section as follow below:

A= management of infection, by mechanical and electrical stimulation.

Methods section

-  Line 115: PICOS tool was used, and it was considered, P (problem). PICO is fine to use for clinical research etc. But I am not sure what you mean by ‘it was considered a problem’.

Answer - We thank the reviewer for the positive remark. In fact, we agree with your suggestion and decided for just use the PICO tool for preclinical studies instead PICOS tool. Thus our "P" became a population and not even more "Patients/ problem" as stated in PICOS tool for clinical studies. We added the following in the methods section:

A= P (Population) - experimental animal models and any method of inducing a wound model.

- Line 163: please re-write that part as it is difficult to understand (The aims of all studies were to assess effects of LIV on wound healing, wounds were induced and after induction, the study started).

Answer - We appreciated your consideration, so we rewrote this sentence in the “ABSTRACT” section as follow below:

A= Likewise, in all studies wounds were induced and after induction, the study started.

Good risk of bias analysis

Discussion

Correctly stated that there were differences in the WBV protocols utilized by the different studies. This is an essential factor when examining the effects of WBV, but also when assessing the safety of the method. The discussion is nicely presented. Please include ALL the in-text citations in brackets [number] and the dates that follow the names of the authors in parentheses.

Answer - We thank the reviewer for the constructive contribution to our manuscript. We added the square brackets around all citations in the text as the dates that follow author names.

Reviewer 3 Report

This manuscript represents the review article with the goal to summarize studies in experimental animal models using whole-body vibration exercise in would healing. These are my comments and suggestions:

Abstract:

Abstract is clear and concise. I advise to better formulate the conclusion - for which population (animals, humans?).

Introduction:

References should be in brackets. Please, describe potential physiological impact of WBW in greater detail.

Methods:

Avoid using future tense in Methods. What was time period of search? From the inception of these databases or the time period was defined, e.g. last ten years etc. Did you register the review? Did you define design of included studies? Did you use platform for reviewing the studies? Did you include/exclude articles written in specific languages?

In general, reformulate your Methods section according to PRISMA guidelines.

Results:

Results chapter is adequate.

Discussion:

References should be in brackets. Better explain why the results could be transferable to humans. Also, why this could be clinically important? Why your review is important and what is the scientific contribution?

Author Response

Manuscript Minor Revision,

Reviewer #3: IJERPH-2210179

Abstract: Abstract is clear and concise. I advise to better formulate the conclusion - for which population (animals, humans?).

Answer - We thank the reviewer for the positive remark, so we rewrote this sentence in the “ABSTRACT” section as follow below:

A= In conclusion, the various biological effects of the response to the WBV exercise indicate the relevance of this intervention in wound healing in animals. Moreover, considering the translation approach, it is possible to speculate that the beneficial effects of this non-pharmacological therapy might justify clinical trials for wound healing also in humans, after criterion evaluation.

Introduction

References should be in brackets. Please, describe potential physiological impact of WBW in greater detail.

Answer - We appreciated your constructive contribution to our manuscript. We added the square brackets around all citations in the text. In addition, we have described the potential physiological impact of WBV as follow below:

A= The potential physiological effects of WBV on different organs/tissues would be related to possible neuromuscular responses and the tonic vibration reflex [19,20,37]. Moreover, the mechano-transduction mechanism in which body cells, once stimulated by the mechanical vibration, modulating the biological activity through specific signaling pathways, as releasing  hormones and other substances (e.g., amino acids, proteins, lipids, ions) [17].

Methods

Avoid using future tense in Methods. What was time period of search? From the inception of these databases or the time period was defined, e.g. last ten years etc. Did you register the review? Did you define design of included studies? Did you use platform for reviewing the studies? Did you include/exclude articles written in specific languages?

In general, reformulate your Methods section according to PRISMA guidelines.

Answer - Answer - We appreciated your consideration. We carefully reviewed the text, and we did not observe future tense (as "will" and "going to", words that form the future tense structure). Moreover, we did not define the period (by year limit) in our search method.

The search about this systematic review in the International Prospective Register of Sys-tematic Reviews (PROSPERO) revealed that there was not review in this subject. The sub-mission protocol of the current systematic in the PROSPERO is 403483.

Furthermore, we did not use platform for reviewing the studies, our design of included studies is defined in METHODS section at INCLUSION CRITERIA topic and we excluded articles written in specific languages. As describe bellow:

A= Exclusion criteria: 7) articles published in other languages (e.g., Spanish, Chinese, Russian, Arabic);

Results

Results chapter is adequate.

Discussion

References should be in brackets. Better explain why the results could be transferable to humans. Also, why this could be clinically important? Why your review is important and what is the scientific contribution?

Answer - We thank the reviewer for the constructive contribution to our manuscript. We added the square brackets around all citations in the text. Also, the response to your questions was writing in the ending of the Discussion section, as shown below:

A= Then, it is possible to endorse, as a fact and scientific contribution, that WBV exercise is useful to treat chronic wounds in animals, and, considering the translational approach, this non-pharmacological intervention would be a perspective for a clinical application in human beings. Moreover, the findings noticed in this review are relevant to support the formulation of appropriate protocols in clinical studies, considering the benefits and biological effects of the WBV exercise.

Reviewer 4 Report

General questions

Congratulations for the nice work. This work presents a good scientific relevance. The discussion includes important topics regarding the physiological mechanisms. However, some issues need to be clarified in addition to other corrections are required. For example, review the numerical order of all citations in the text and insert parentheses.

Major questions

- keywords: Please, correct words to increase manuscript search in database. If the word is in the title, it doesn't have to be in the keywords.

- Please, put parentheses or square brackets around all citations in the text (Line 48: annually1)

- Line 115: “PICOS tool was used...” Who are the patients/sample? (in the text)

- Line 36: “From 55 studies, 5 studies met the inclusion criteria” and Line 142 “According to the search strategy performed in electronic databases, a total of 48 arti...” and Figure 1 (54 articles: 48 + 4). Which of these three statements is correct?

- Line 145: “...33 articles were screened, and the abstracts reviewed. Of those, 28 articles were excluded because the abstracts showed that they did not match the inclusion criteria. Thus, 7 articles were identified as potentially eligible and subjected to more detailed analysis”. However, 33-28 is 5 not 7. (See also Figure 1).

- Line 160: “...large number of studies...”, 3 does not represent a large number of studies, but may represent a large proportion (please adjust).

- Line 292: reference 56 is quoted in the introduction (Line 79). Something is wrong because it is the last reference on the list. Authors need to review the entire numerical order of references in the text.

Minor questions

- Line 34: November 21rd

- Line 292: 56. Mon ami, please, add this reference here... Cunha de Sá-Caputo D, Seixas A, Taiar R, Bernardo-Filho M. Vibration Therapy for Health Promotion. Complementary Therapies [Internet]. 2022 Jul 6; Available from: http://dx.doi.org/10.5772/intechopen.105024 (This comment shouldn't be here).

Author Response

Manuscript Minor Revision, Reviewer #4: IJERPH-2210179

Major questions

- keywords: Please, correct words to increase manuscript search in database. If the word is in the title, it doesn't have to be in the keywords.

Answer - We thank the reviewer for the positive remark. We added the following in the keywords section:

A= Keywords: vibration therapy; animal studies; wound, healing, skin

- Please, put parentheses or square brackets around all citations in the text ( Line 48: annually1)

Answer - We thank the reviewer for the constructive contribution to our manuscript. We added the square brackets around all citations in the text.

- Line 115: “ PICOS tool was used...” Who are the patients/sample? (in the text)

Answer - We thank the reviewer for the constructive remark. Thus, we decided for just use the PICO tool for preclinical studies instead PICOS tool. Thus our "P" became a population and not even more "Patients/ problem" as stated in PICOS tool for clinical studies. We added the following in the methods section:

A= P (Population) - experimental animal models and any method of inducing a wound model.

- Line 36: “ From 55 studies, 5 studies met the inclusion criteria” and Line 142 “According to the search strategy performed in electronic databases, a total of 48 arti...” and Figure 1 (54 articles: 48 + 4). Which of these three statements is correct?

Answer - We thank the reviewer for the positive observation regard the calculation in the Figure 1. We did the corrections, and this section was refreshed.

- Line 145: “...33 articles were screened, and the abstracts reviewed. Of those, 28 articles were excluded because the abstracts showed that they did not match the inclusion criteria. Thus, 7 articles were identified as potentially eligible and subjected to more detailed analysis”. However, 33-28 is 5 not 7. (See also Figure 1).

Answer - In the same way, we did the corrections in the “SEARCH RESULTS” paragraph and this section was refreshed.

- Line 160: “...large number of studies...”, 3 does not represent a large number of studies, but may represent a large proportion(please adjust).

Answer - We appreciate your consideration, so we rewrote this sentence in the first paragraph of the topic “SELECTED STUDIES” as follow below:

A= a large percentage of studies (60%, n = 3).

- Line 292: reference 56 is quoted in the introduction (Line 79). Something is wrong because it is the last reference on the list. Authors need to review the entire numerical order of references in the text.

Answer - We thank the reviewer for the remark. We did the adjustment in this reference.

Minor questions

- Line 34: November 21rd

- Line 292: 56.

Answer - Once upon time, we thank the reviewer for the positive remarks and the constructive contributions to our manuscript. Considering your minor questions, we did the adjustments according to your suggestions.

Round 2

Reviewer 1 Report

The authors have improved the quality of the manuscript with your corrections.

Author Response

REVIEWER 1

Author's Reply to the Review Report (Reviewer 1)

Please provide a point-by-point response to the reviewer’s comments and either enter it in the box below or upload it as a Word/PDF file. Please write down "Please see the attachment." in the box if you only upload an attachment. An example can be found here.

* Author's Notes to Reviewer

p

Word / PDF

or

Haut du formulaire

Review Report Form

Open Review

Quality of English Language

( ) English very difficult to understand/incomprehensible
( ) Extensive editing of English language and style required
( ) Moderate English changes required
(x) English language and style are fine/minor spell check required
( ) I am not qualified to assess the quality of English in this paper

Yes

Can be improved

Must be improved

Not applicable

Does the introduction provide sufficient background and include all relevant references?

(x)

( )

( )

( )

Are all the cited references relevant to the research?

(x)

( )

( )

( )

Is the research design appropriate?

(x)

( )

( )

( )

Are the methods adequately described?

(x)

( )

( )

( )

Are the results clearly presented?

(x)

( )

( )

( )

Are the conclusions supported by the results?

(x)

( )

( )

( )

Comments and Suggestions for Authors

The authors have improved the quality of the manuscript with your corrections.

We thank this reviewer for the comments/questions. The scientific quality of the manuscript was improved.

Submission Date

25 January 2023

Date of this review

06 Mar 2023 18:36:15

Bas du formulaire

Reviewer 4 Report

Major questions

- Line 36: “From 55 studies, 5 studies met the inclusion criteria” and Line 142 “According to the search strategy performed in electronic databases, a total of 48 arti...” The problem of the difference in the number of studies continues: see abstract (line 36) and Line 155, please. Which of these statements are correct?

Author Response

Reviewer 4

Review Report Form

Open Review

Quality of English Language

( ) English very difficult to understand/incomprehensible
( ) Extensive editing of English language and style required
( ) Moderate English changes required
(x) English language and style are fine/minor spell check required
( ) I am not qualified to assess the quality of English in this paper

Yes

Can be improved

Must be improved

Not applicable

Does the introduction provide sufficient background and include all relevant references?

(x)

( )

( )

( )

Are all the cited references relevant to the research?

(x)

( )

( )

( )

Is the research design appropriate?

(x)

( )

( )

( )

Are the methods adequately described?

(x)

( )

( )

( )

Are the results clearly presented?

(x)

( )

( )

( )

Are the conclusions supported by the results?

(x)

( )

( )

( )

Comments and Suggestions for Authors

Major questions

- Line 36: “From 55 studies, 5 studies met the inclusion criteria” and Line 142 “According to the search strategy performed in electronic databases, a total of 48 arti...” The problem of the difference in the number of studies continues: see abstract (line 36) and Line 155, please. Which of these statements are correct?

Thank you for your comments. It was only wrong in the abstract (line 35)….it was …. From 55 studies, 5 studies met the inclusion criteria”….and it was corrected to… « From 48 articles, 5 studies met the inclusion criteria”…

We thank this reviewer for the comments/questions. The scientific quality of the manuscript was improved.

Submission Date

25 January 2023

Date of this review

06 Mar 2023 02:23:42
